# Two-Dimensional MXene as a Promising Adsorbent for Trihalomethanes Removal: A Density-Functional Theory Study

**DOI:** 10.3390/nano14050454

**Published:** 2024-02-29

**Authors:** Islam Gomaa, Nasser Mohammed Hosny, Hanan Elhaes, Hend A. Ezzat, Maryam G. Elmahgary, Medhat A. Ibrahim

**Affiliations:** 1Nanotechnology Research Centre (NTRC), The British University in Egypt (BUE), Suez Desert Road, El-Sherouk 11837, Egypt; islam.gomaa@bue.edu.eg; 2Department of Chemistry, Faculty of Science, Port Said University, Port Said 42522, Egypt; nasserh56@yahoo.com; 3Physics Department, Faculty of Women for Arts, Science and Education, Ain Shams University, Cairo 11757, Egypt; medahmed@yahoo.com; 4Nano Unit, Space Lab, Solar and Space Research Department, National Research Institute of Astronomy and Geophysics (NRIAG), Helwan 11421, Egypt; hend.ezzat@nriag.sci.eg; 5Chemical Engineering Department, The British University in Egypt (BUE), El Sherouk 11837, Egypt; 6Department of Chemical Engineering, Massachusetts Institute of Technology, 77 Massachusetts Avenue, Cambridge, MA 02139, USA; 7Spectroscopy Department, National Research Centre, 33 El-Bohouth St., Dokki 12622, Egypt; medahmed6@yahoo.com; 8Molecular Spectroscopy and Modeling Laboratory, Centre of Excellence for Advanced Science, National Research Centre, 33 El-Bohouth St., Dokki 12622, Egypt

**Keywords:** halomethanes (THs), MXene (M_n+1_·X_n_T_x_), DFT, TDM, MESP

## Abstract

This groundbreaking research delves into the intricate molecular interactions between MXene and trihalomethanes (THs) through a comprehensive theoretical study employing density-functional theory (DFT). Trihalomethanes are common carcinogenic chlorination byproducts found in water sanitation systems. This study focuses on a pristine MXene [M_n+1_·X_n_] monolayer and its various terminal [T_x_] functional groups [M_n+1_·X_n_T_x_], strategically placed on the surface for enhanced performance. Our investigation involves a detailed analysis of the adsorption energies of THs on different MXene types, with the MXene-Cl layer emerging as the most compatible variant. This specific MXene-Cl layer exhibits remarkable properties, including a total dipole moment (TDM) of 12.443 Debye and a bandgap of 0.570 eV, achieved through meticulous geometry optimization and computational techniques. Notably, THs such as trichloromethane (CHCl_3_), bromide-chloromethane (CHBrCl_2_), and dibromochloromethane (CHBr_2_Cl) demonstrate the highest TDM values, indicating substantial changes in electronic and optical parameters, with TDM values of 16.363, 15.998, and 16.017 Debye, respectively. These findings highlight the potential of the MXene-Cl layer as an effective adsorbent and detector for CHF_3_, CHClF_2_, CHCl_3_, CHBrCl_2_, and CHBr_2_Cl. Additionally, we observe a proportional increase in the TDM and bandgap energy, indicative of conductivity, for various termination atom combinations, such as Mxene-O-OH, Mxene-O-F, Mxene-O-Cl, Mxene-OH-F, Mxene-F-Cl, and Mxene-OH-Cl, with bandgap energies measured at 0.734, 0.940, 1.120, 0.835, and 0.927 eV, respectively. Utilizing DFT, we elucidate the adsorption energies of THs on different MXene surfaces. Our results conclusively demonstrate the significant influence of the termination atom nature and quantity on MXene’s primitive TDM value. This research contributes to our understanding of MXene–THs interactions, offering promising avenues for the development of efficient adsorbents and detectors for THs. Ultimately, these advancements hold the potential to revolutionize water sanitation practices and enhance environmental safety.

## 1. Introduction

Due to the widespread use of water disinfection to eliminate disease-causing bacteria in the distribution system, humans are constantly exposed to the byproducts of disinfection through ingestion, inhalation, and skin absorption during routine water usage and consumption [1]. Disinfectants react with organic matter in source waters to produce a family of chemicals known as disinfection byproducts [2,3]. Chlorination is the most efficient and affordable form of disinfection. It is frequently used to treat tap water and swimming pool water [4]. However, chlorination may generate a variety of byproducts of chlorine that have the potential to cause cancer, particularly halogenated organic byproducts such as trihalomethanes (THMs), haloacetic acids (HAAs), and others [5,6]. THMs, which make up 66% of all disinfection byproduct chemicals, are the most prevalent species in chlorinated water. Blood concentration measurements are sensitive to very low levels of exposure and serve as integrative measurements of exposure from many pathways [7]. Due to the frequent daily water-use activities and the slower partitioning out of adipose tissue, blood THM concentrations are expected to be generally stable [8,9]. The risk of cancer in child swimmers from exposure to THMs is critical and scary, according to recent statics [10,11,12]. This risk can be deemed unsatisfactory and requires improvement. Attendance at chlorinated swimming pools has been linked to an increased risk of asthma and allergy illnesses in kids and teenagers, according to a number of population studies [13,14]. The majority of earlier studies calculated disinfection byproduct exposures based on questionnaire- or environmental-monitoring-reported swimming frequency, which was prone to exposure misclassification because it did not take into account variables that might affect individual intake and disinfection byproduct metabolism [15]. THM removal and sensing applications lack scientific research or practical foundation due to their relatively short lifetime and need more in-depth knowledge to obtain the optimum choice that can be used in in situ capturing and sensing. A smart sensor is able to convert the physical state of an object or environment, such as temperature, light, sound, and/or motion, into electrical or other types of signals that may be further processed [16,17]. A single smart sensor node may have several sensors measuring different physical quantities. Transition metal carbides, nitrides, and carbonitrides, known as MXenes, stand out as a novel class of 2D materials with a distinctive metallic behavior beyond graphene and other 2D semiconducting materials [18,19]. The exciting discovery of MXene nanomaterials has had a significant impact on the field of sensors for a variety of applications, including wearable health sensors and gas sensors [20,21,22]. The molecular structure chemistry of MXene has the formula M_n+1_X_n_T_x_, where M is the transition metal core, X is C or N, and T_x_ indicates the surface terminations (O, OH, Cl, and F) [23]. The n value can range from 1 to 4; two or more transition metal atoms can occupy the M sites, resulting in solid solutions or arranged structures [24]. In the experimental spectrum, the exfoliated primitive 2D Ti_3_C_2_ layers possess two exposed Ti atoms per unit formula that should be surrounded by suitable ligands. The building block of MXene, M_n+1_X_n_T_x_, is terminated by various functional groups depending on the structure of the MAX phase precursor and the etching method [25]. The manipulation of the aqueous synthesis environment, including the nature of the solvent and the concentration of ions, enables the preferential formation of specific terminations during MXene synthesis [24,26]. Surface plasmon resonance (SPR) is an optical method used to study molecule interactions in real time [27]. SPR can be carried out when plane-polarized light strikes a metal film under total internal reflection circumstances, which may be next step for our work. Modeling of each molecule was conducted by attaching respective ligands to the exposed Ti atoms followed by full-geometry optimizations according to the real position in the literature [28]. Two-dimensional quantum dots (2D-QDs) are nanoscale semiconductor particles confined to two dimensions, exhibiting quantum confinement effects that strongly influence their electronic and optical properties [29]. Unlike traditional three-dimensional quantum dots, the dimensionality of 2D-QDs is restricted to two spatial dimensions within 1–10 nm. These nanocrystals, often composed of materials like cadmium selenide or lead sulfide, hold promise for applications in electronics, photonics, and optoelectronics. Their unique size-dependent properties make them suitable for use in devices such as solar cells, light-emitting diodes (LEDs), and sensors. Researchers are actively exploring various synthesis methods and characterizing the properties of 2D-QDs to harness their potential in technological advancements [30,31]. Unfortunately, 2D-QDs such as Mxene need sophisticated conditions like laser pulse irradiation [32,33]. An MXene (Ti3C2) QD-monolayer was utilized in this study to investigate the influence of MXene functionalization with different surface ligands (hydroxyl, fluorine, oxygen, chlorine, and mixed-ligand ions) on the characteristic electronic properties of MXene using DFT calculations. The electronic properties and reactivity of primitive MXene and functionalized MXene were investigated in terms of TDM, HOMO/LUMO band gap (ΔE), and MESP to attain the optimum functionalized form compatible with THs sensitivity and accuracy. To the best of our knowledge, there has been no research work reported on TH detection by pristine MXenes and ligand-terminated MXenes. Also, no specific studies have been conducted on TH molecules adsorption on 2D monolayer MXene surfaces as a brief explanation of MESP, ΔE, and TDM between the adsorbed TH molecules and the substrate. Therefore, we took the closest reliable structure that would offer a hands-to-ligand selection trend based on the specific detection and selectivity requirements in further experimental and theoretical works.

## 2. Calculation Details

The structural models representing MXene (Ti_3_C_2_) and its functionalized variants with different ligands (hydroxyl, fluorine, oxygen, chlorine, and mixed-ligand ions) underwent optimization using the Gaussian09 software package (Gaussian, Inc., Wallingford, CT, USA) [34]. The calculations were conducted at the Molecular Spectroscopy and Modeling unit, National Research Centre, Egypt. Optimization of the simulated structures employed density-functional theory (DFT) at the B3LYP level, utilizing the LANL2DZ basis set [35,36,37]. A comprehensive elucidation of the computational details is provided in the Appendix A. Each model was built using Gauss View 05 using the program’s default parameters, and then optimization was conducted with the proposed computational method. The calculated physical parameters were conducted upon the optimized structures. The output of the calculated parameters was also aided by the Gauss View 05 program. TDM was estimated directly, while the band gap energy ∆E was estimated as the energy difference between the HOMO and LUMO. Ads energy E_Ads_ was calculated as E_Ads_ = E_Total_ − [E_MXene-Cl_ + E_Ad. mol_.]. All calculations adhered to the default convergence criteria in Gaussian 09, ensuring consistency with established methodologies [38,39,40].

## 3. Results and Discussion

### 3.1. Building Model Molecule

MXene’s chemical structure is based on the formula M_n+1_X_n_T_x_, where M represents the transition metal including Ti, V, Zr, or Nb, X represents C and/or N elements, and T represents functionalizing groups such as O, OH, Cl, and F. Here, we present a comprehensive theoretical study of a 2D MXene-like structure (Ti_3_C_2_) monolayer [41,42] through a carbonized Ti_3_C_2_ structure terminated with different functional groups like hydroxyl (OH), halide (Cl and F), and oxygen (O) ions [43], as well as mixed-ion termination phases. Such a monolayer will not only improve the chemical stability but will also have different favorable properties due to the change in the binding mode, with the carbon atoms forming C_2_ dimers in the Ti_3_C_2_ monolayer [44]. Figure 1a shows the optimized structure of the MXene (Ti_3_C_2_) monolayer. Figure 1b–k show the functionalized MXene with different functional groups (O, OH, Cl, and F) and mixed-group termination phases. The choice of specific models and surface coverages in this manuscript stemmed from a deliberate and strategic approach aimed at balancing computational feasibility with the need for an accurate representation of MXene structures and their interactions with functional groups. In addressing the issue of computational tractability, we opted for a surface coverage of n = 5, strategically balancing computational efficiency with the need to accurately represent the MXene structures; a larger surface coverage could potentially compromise the feasibility of the study due to increased computational demands. To overcome the computational constraints, we employed representative sampling, selecting n = 5 as a compromise that still provided insightful and representative observations into the electronic and structural modifications induced by the various functional groups. This choice ensured that our models could capture the essential features without sacrificing the overall computational efficiency. Furthermore, in exploring the diversity of functionalization patterns, we incorporated a variety of functional groups (T=F, Cl, O, OH) in combinations such as MXene-10O, MXene-10OH, MXene-10F, etc. This deliberate inclusion allowed for a comprehensive investigation into the different functionalization patterns and compositions, facilitating a nuanced understanding of how the diverse functional groups impact the MXene structure.

### 3.2. Total Dipole Moment and HOMO/LUMO Bandgap Energy

Table 1 provides the computed TDM and ΔE values for the MXene and functionalized MXene terminated with different functional groups such as O, OH, Cl, and F as well as their mixed-ion termination ratio. The model molecules attached with the primitive proposed structure of MXene were computed using the B3LYP/LANL2DZ model. The TDM of MXene, which was 6.399 Debye, decreased in the case of O termination to 5.153 Debye with increase in ΔE from 0.633 to 0.766 eV. The TDM increased to 10.927 Debye with an increase in E to 0.855 eV when MXene was functionalized with an -O atom. Simultaneously, MXene functionalization with the -OH group increased the TDM to 13.019 Debye and E to 1.486 eV. The TDM improved for both halides, increasing to 18.386 and 12.443 Debye, respectively, while E increased for -F and decreased for -Cl, increasing to 1.125 and 0.570 eV, respectively. However, in all of the proposed structural model molecules of MXene functionalized with mixed ions, the TDM increased for MXene-O-OH, MXene-O-F, MXene-O-Cl, MXene-OH-F, MXene-OH-Cl, and MXene-F-Cl to 12.024, 13.213, 10.792, 18.018, 7.379, and 13.569 Debye, respectively, in addition to an increase in ΔE to 0.734, 0.940, 0.740, 1.120, 0.835, and 0.927 eV, respectively. Based on the computed results, it appears that the TDM value of primitive MXene increased and changed according to the nature and number of the termination atoms [30]. The geometry optimization and calculation of the chlorinated structure was observed to be the most reactive and stable one, with its TDM measuring 12.443 Debye and its ΔE decreasing to 0.570 eV [45]. The following are the proposed explanations for the changes in the total dipole moment (TDM) and energy gap (E) of MXene after functionalization with various atoms and groups. Functionalization with -O atoms: The addition of -O atoms to the MXene surface resulted in the formation of new electron-donating functional groups. This raised the overall electron density of the MXene surface, thereby raising the TDM. The increase in E was caused by the destabilization of MXene’s Fermi level during the -O functionalization. Functionalization with –OH groups: The addition of –OH groups to the MXene surface introduced new polar functional groups. This also increased the TDM of the MXene. The increase in ΔE was more significant than in the case of –O functionalization due to the stronger electron-donating ability of the –OH group. Functionalization with–F and –Cl halides: The addition of halide atoms to the MXene surface introduced new electron-withdrawing functional groups. This decreased the overall electron density of the MXene surface, which in turn decreased the TDM. The decrease in ΔE for –F functionalization was due to the stabilization of the Fermi level of MXene upon –F functionalization. The increase in ΔE for –Cl functionalization was due to the destabilization of the Fermi level of MXene upon –Cl functionalization. Functionalization with mixed ions: The functionalization of MXene with mixed ions resulted in a complex interplay of electron-donating and electron-withdrawing effects. The overall TDM and ΔE of the functionalized MXene depended on the relative strengths of these competing effects. In general, the TDM of MXene was increased by functionalization with electron-donating groups, such as –O and –OH. The TDM of MXene was decreased by functionalization with electron-withdrawing groups, such as –F and –Cl. The ΔE of MXene was increased by functionalization with groups that destabilized the Fermi level, such as –O and –OH. The ΔE of MXene was decreased by functionalization with groups that stabilized the Fermi level, such as –F. The fact that the chlorinated MXene structure was the most reactive and stable was most likely due to a combination of factors, including its increased TDM and decreased E. The increased TDM made the chlorinated MXene surface more appealing to adsorbates. The decreased E made the chlorinated MXene surface more reactive to chemical reactions.

### 3.3. Molecular Electrostatic Potential (MESP)

MESP maps were generated for the proposed chemical structures as presented in Figure 2. The MESP maps were studied at the DFT theoretical level by utilizing the B3LYP/LANL2DZ model. MESP maps, in general, offer a straightforward method of elucidating the distribution of electronic charges within a structure under study, thus pinpointing the most probable active centers. Red, orange, yellow, green, cyan, and dark blue are the hues that make up the MESP color spectrum, which goes from the most negative to the most positive. Red is associated with a lower negative potential, whereas blue is associated with a positive potential. Also, the orange region indicates a lower negative potential than red, and yellow indicates the neutral potential region [46]. The calculated maps, as presented in Figure 2, consisted of only two colors, red and yellow, representing the extreme negative region for the red color and neutral sites for yellow. The distribution of the charges of the atoms can also be related to some extent via the electronegativity of the atoms to which they are attached. Atoms with a high electronegativity are surrounded by red color when combined with atoms other than electronegative ones. The presence of several atoms with a nearly equal electronegativity narrows the color distribution considerably. As a result, MESP maps can be used to determine whether an active site of interest can undergo a chemical interaction. MESP is important because it can show how the overall charge distribution affects the dipole moment, electronegativity, partial charges, and chemical reactivity location of a structure. Regarding the MESP map of the primitive MXene layer, there was only yellow color overlap within the layer, while a red color surrounded the MXene terminals, indicating that the MXene layer’s surface had a low activity, with the activity concentrated around the layer edges. As presented in Figure 2a. Therefore, the TDM and ΔE results were confirmed by the MESP map results. However, due to doping with different terminal ions, some changes were observed, as shown in Figure 2. Figure 2b–k depict the increase in the electronegativity of MXene’s structure upon its interaction with O, OH, F, Cl, and mixed ions. The red color was located around the MXene layer’s surface and extended to some regions within O. This means that the electronegativity of the MXene layer’s surface increased due to the doping with different functional groups (–O, −OH, –Cl, and –F), especially the chlorine ions. Furthermore, when the intensity of the red hue was increased, adding more ligands led to an increase in the tested models’ electronegativity. The TDM results and ΔE were confirmed by the increase in the red color intensity when the quantity of doping ligands was increased. This indicates that the surface of the MXene layer was more reactive and could be utilized for sensing purposes. It is worth mentioning that the functionalization of MXene with different atoms and groups can be used to modulate its electronic properties, including its TDM, energy gap (ΔE), and MESP. This is because different functional groups have different electron-donating or electron-withdrawing abilities. For instance, the TDM and ΔE of the MXene increased when electron-donating groups, like -O and -OH, were added to the surface, whereas the TDM and ΔE of the MXene dropped when electron-withdrawing groups, like -F and -Cl, were added. The MESP maps of MXene can be used to visualize the distribution of the electronic charges on the MXene surface and how this distribution is affected by functionalization. For example, the functionalization of the MXene with electron-donating groups increased the electronegativity of the MXene surface, while the functionalization of the MXene with electron-withdrawing groups decreased the electronegativity of the MXene surface. Accordingly, the MESP maps coincided with the TDM and band gap results, which confirmed that the enhancement occurred according to functionalization.

### 3.4. MXene-Cl for Trihalomethanes Adsorption

#### 3.4.1. Trihalomethanes Model Structure

THMs, which are the most prevalent byproducts of water chlorination, are chemical compounds in which three hydrogen atoms of methane (CH_4_) are replaced by halogen atoms such as Cl, Br, and I. Common THMs are found in higher levels than other organohalogen contaminants, which are environmental hazards, most of which are carcinogenic [46]. THMs are formed when chlorine combines with a range of organic molecules in water, including natural organic matter on the water surface such as humic acid and fulvic acid, as well as chemicals discharged by industrial wastewater, agricultural drainage, and solid waste leachates [47]. Accordingly, a model molecule of functionalized MXene-Cl was used to study the possibility of using it for detecting common THMs. A model of a functionalized MXene-Cl sheet for CHCl_3_ probed through different MXene-Cl active sites is shown in Figure 3.

#### 3.4.2. Trihalomethane Adsorption on Functionalized MXene-Cl Sheet

The adsorbing energies of the functionalized MXene-Cl sheet for CHCl_3_ sensing through MXene-Cl at different active sites were calculated. The total energy and adsorbing energies were estimated and are presented in Table 2 using the adsorption energy from the following formula:*E*_Ads_ = *E*_Total_ – [*E*_MXene-Cl_ + *E*_Ad. mol._](1)

E_Ads_ is the binding energy of the adsorbed THM atom (CHCl_3_) with different functionalized MXene sheets with Cl. E_MXene-Cl_ is the total energy of one sheet of the MXene-Cl, and E_Ad. mol._ Is the total energy of an individual molecule of the adsorbed molecule (Ad. Mol.) of THM (CHCl_3_) [42]. The CHCl_3_ adsorption energy on the MXene-Cl through Ti atoms on the surface was negative, suggesting that the adsorption structure was stable and that the adsorption process was exothermic, as shown in the data in Table 2. When the calculated values were compared, the best CHCl_3_ adsorption site on the MXene-Cl surface was found to be through Ti atoms, and this was consistent with previous works [48,49], as it demonstrated the lowest required binding energy and also resulted in the most stable structure, as evidenced by it having the lowest total energy. Notably, the exothermic nature of the adsorption process and the stability of the adsorption structure are indicated by the negative adsorption energy. This was because when the adsorbate and adsorbent are bonded together, the system is more stable than when they are free. The adsorption process is exothermic, meaning that heat is emitted as a byproduct of energy release. Ti atoms on the surface provided the lowest necessary binding energy for CHCl_3_ adsorption on the MXene-Cl. This was because the Ti atoms on the MXene surface had a greater affinity for CHCl_3_ molecules than other atoms. This was most likely because Ti atoms are more appealing to the electronegative chlorine atoms in CHCl_3_ molecules due to their compatible electronegativity compared to other atoms on the MXene surface. The Ti atoms on the surface provided the most stable structure for CHCl_3_ adsorption on the MXene-Cl.

#### 3.4.3. Functionalized MXene-Cl for Common Trihalomethane Sensing

As a result, MXene-Cl was adopted to examine the detection of common THMs via the probable and appropriate position of connection through the Ti atoms, as illustrated in Figure 4. To examine the detection and selectivity of the MXene-Cl layer of common THMs (CHF_3_, CHClF_2_, CHCl_3_, CHBrCl_2_, CHBr_2_Cl, CHBr_3_, and CHI_3_), the TDM and ΔE of MXene-Cl’s interactions with common trihalomethanes were estimated, as shown in Table 3. The TDM of all the structures of MXene-Cl with common THMs (CHF_3_, CHClF_2_, CHCl_3_, CHBrCl_2_, CHBr_2_Cl, CHBr_3_, and CHI_3_) was increased, except for CHBr_3_ and CHI_3_. As a result, the MXene-Cl layer could operate as a sensor for CHF_3_, CHClF_2_, CHCl_3_, CHBrCl_2_, and CHBr_2_Cl and could be particularly selective for CHCl_3_, CHBrCl_2_, and CHBr_2_Cl, which had the highest TDM values of 16.363, 15.998, and 16.017 Debye, respectively [50]. Remarkably, considering the multiple physical, chemical, and electronic properties, we can suggest the following main reasons for these results. Surface chemistry: The MXene-Cl layer possesses a unique surface chemistry due to the presence of functionalized chlorine atoms. These chlorine atoms can interact with the common THMs through chemical bonding or adsorption mechanisms. The surface chemistry of MXene-Cl plays a crucial role in facilitating the sensor’s behavior. Intermolecular interactions: The MXene-Cl layer can engage in intermolecular interactions with the common THMs, such as van der Waals forces, dipole–dipole interactions, or even hydrogen bonding. These interactions arise from the polar nature of the THMs and the ability of MXene-Cl to accommodate and stabilize these molecules. Total dipole moment (TDM): The TDM is a measure of the difference in the electric charge distribution in a molecule, indicating the strength and polarity of its dipole. The MXene-Cl layer’s interaction with the common THMs leads to changes in the TDM values, suggesting alterations in the electronic charge distribution and dipole moments of the THMs. Polarizability and dielectric properties: The MXene-Cl, with its layered structure and specific electronic properties, can influence the polarizability and dielectric properties of the system. The presence of the MXene-Cl layer enhances the ability of the THMs to induce dipoles within the surrounding medium, leading to changes in the overall dielectric response. Band alignment and charge transfer: The MXene-Cl possesses a specific band structure and electronic properties. When in contact with the common THMs, there could be a modulation of the band alignment and charge transfer phenomena. Such charge transfer processes influence the TDM values, indicating the involvement of electronic interactions between the MXene-Cl layer and the THMs. In contrast to selective binding sites, the MXene-Cl may offer specific binding sites or active sites that favor the interaction with certain THMs over others. The molecular structure and surface features of the MXene-Cl layer contribute to its selective affinity towards CHCl_3_, CHBrCl_2_, and CHBr_2_Cl, as evidenced by their higher TDM values. This theoretical investigation has unveiled the immense potential of MXene-based materials for a wide spectrum of real-world applications. These innovative materials hold promise for the development of highly sensitive sensors for trihalomethanes (THMs) in water, owing to the exceptional TDM values of the MXene-Cl layer that enable the detection of THMs at ultra-low concentrations, as inspired by previous work [51,52,53]. Additionally, the robust chemisorption of THMs onto MXene surfaces suggests their potential as catalysts for THM degradation in water. Their remarkable surface area and catalytic activity make them eminently suitable for this application. MXene-based materials could also be harnessed to develop novel materials for water purification, leveraging their exceptional surface area and adsorption capacity to effectively remove contaminants from water.

Despite the immense promise of MXene-based materials, their translation into real-world applications faces several formidable challenges. The synthesis of MXenes is a complex and multi-step process, posing significant hurdles for large-scale production. Moreover, MXenes are susceptible to oxidation and degradation in air and water, potentially limiting their long-term performance and durability. Controlling the functionalization of MXenes with diverse terminal groups remains a challenge, potentially hindering the development of MXene-based materials with tailored properties for specific applications. Integrating MXene-based materials into functional devices proves challenging due to their high surface area and propensity to restack, limiting the development of sensors, catalysts, and other devices that rely on controlled material architectures. Finally, the environmental impact of MXene production and disposal remains incompletely understood, potentially hindering their widespread adoption and sustainability. Despite these challenges, significant progress is needed in both the theoretical and experimental domains to address these limitations and pave the way for the development of next-generation MXene-based materials with the potential to revolutionize various applications, including water purification, sensing, and catalysis.

This detailed arrangement allows for a nuanced examination of how the MXene-Cl structure responds to different common trihalomethanes, providing valuable insights into the potential applications of functionalized MXene in the sensing of these compounds. The potential applications of these findings can be summarized as follows. Efficient THM removal: This study identified the MXene-Cl layer as a particularly effective variant for adsorbing THMs. This specific layer exhibits remarkable properties, such as a total dipole moment (TDM) and a bandgap, which were achieved through geometry optimization. The compatibility of MXene-Cl with THMs, as demonstrated by its high TDM values for specific compounds, suggests its potential use as an efficient adsorbent for THM removal in water treatment processes. Selective detection: The observed changes in the electronic and optical parameters, including the TDM values, for different termination atom combinations indicate MXene’s potential as a selective detector for specific THMs. This study’s detailed analysis of various termination atom combinations, such as Mxene-O-OH, Mxene-O-F, Mxene-O-Cl, Mxene-OH-F, Mxene-F-Cl, and Mxene-OH-Cl, provides insights into tailoring MXene for selective THM detection. Tailoring material properties: This investigation into the nature and quantity of the termination atoms on MXene’s primitive TDM value contributes to the understanding of how the material properties can be tailored for specific applications. This knowledge could guide the engineering of MXene-based adsorbents with optimized properties for enhanced water treatment efficiency. Revolutionizing water sanitation practices: By providing insights into the molecular interactions between MXene and THMs, this study contributes to the development of advanced materials for water sanitation. MXene’s potential as an adsorbent and detector has implications for revolutionizing current water sanitation practices, offering more efficient and selective approaches for THM removal and monitoring. In conclusion, the results of this study pave the way for the development of MXene-based materials with tailored properties for effective THM removal and detection. These advancements hold great promise for improving water treatment processes, enhancing environmental safety, and addressing the challenges associated with the presence of carcinogenic chlorination byproducts in water systems.

## 4. Conclusions

In summary, this work established a systematic study into the interaction of THMs by analyzing MXene (Ti_3_C_2_) monolayers and their functionalization with different terminal functional groups (–O, −OH, –Cl, and –F) and mixed terminal functional groups by employing a first-principles calculation. For the MXene-O-OH, MXene-O-F, MXene-O-Cl, MXene-OH-F, MXene-F-Cl, and MXene-OH-Cl, the TDM increased, and the ΔE value increased to 0.734, 0.940, 1.120, 0.835, and 0.927 eV, respectively. The calculated results support the hypothesis that the nature and number of the terminal atoms had a fluctuating and expanding impact on the TDM value of the primitive MXene. The adsorption energies of THMs on the surfaces of the various MXene types were calculated using DFT. Through geometry optimization and calculations, the most reactive and stable chlorinated structure, with a TDM of 12.443 Debye and a ΔE of 0.570 eV, was created. The MXene-Cl layer could serve as a sensor for CHF_3_, CHClF_2_, CHCl_3_, CHBrCl_2_, and CHBr_2_Cl because it showed the highest TDM values of 16.363, 15.998, and 16.017 Debye, respectively, and it needs a proper circuit design for its application. Based on the calculated adsorption energies, there was a strong and spontaneous chemisorption of THMs on the chlorinated form (MXene-Cl) with a remarkable change in the TDM and ΔE. The results of our study hold practical significance for water sanitation systems. The identification of the MXene-Cl layer as a highly effective adsorbent for trihalomethanes (THs) suggests potential applications in improving water treatment processes by efficiently removing carcinogenic chlorination byproducts. Additionally, the unique properties of the MXene-Cl layer, such as the total dipole moment (TDM) and bandgap, open avenues for developing sensitive detectors and enhancing the real-time monitoring of THs in water. The knowledge gained on the influence of the termination atoms allows for tailoring MXene-based adsorbents to optimize their performance in specific water treatment applications. Implementing MXene-based materials in water treatment plants has the potential to revolutionize current practices, contributing to more efficient contaminant removal and environmental safety goals. Overall, our findings offer practical solutions for advancing water sanitation practices through the utilization of MXene-based materials.

## Figures and Tables

**Figure 1 nanomaterials-14-00454-f001:**
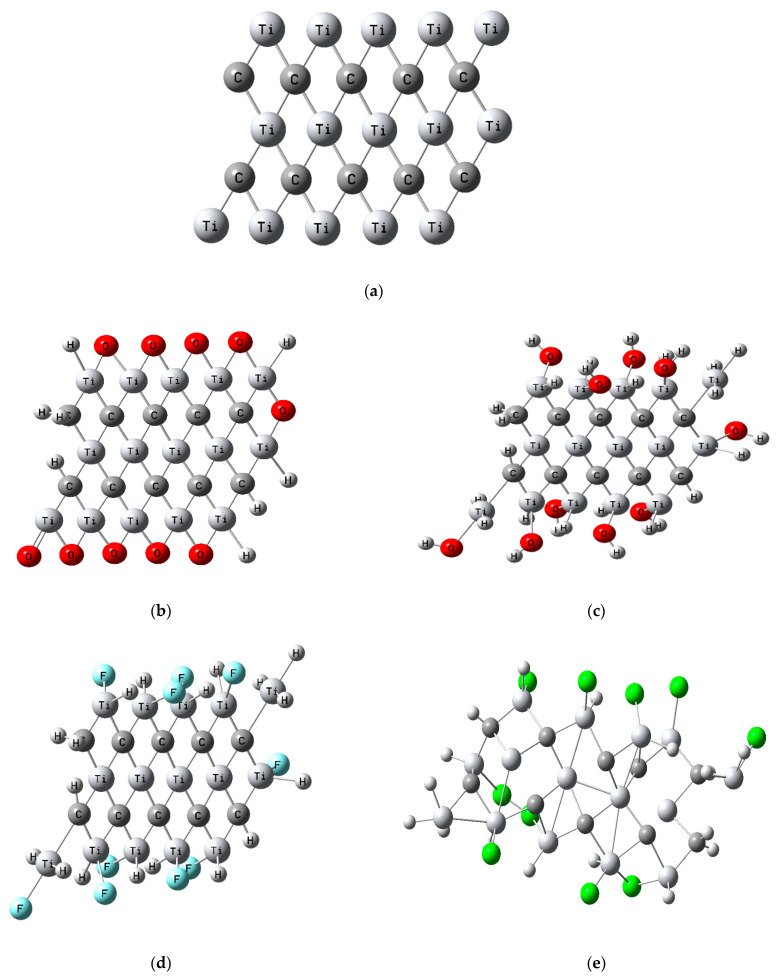
A series of simulated and optimized model structures depicting MXene (Ti_3_C_2_n__) and its functionalized counterpart (Ti_3_C_2_n__T_2_n__) with diverse functional groups (T=F, Cl, O, OH), which are presented as follows: (**a**) pristine MXene, (**b**) MXene-10O (10 oxygen atoms), (**c**) MXene-10OH (10 hydroxyl groups), (**d**) MXene-10F (10 fluorine atoms), (**e**) MXene-10Cl (10 chlorine atoms), (**f**) MXene-5O-5OH (5 oxygen and 5 hydroxyl groups), (**g**) MXene-5O-5F (5 oxygen and fluorine atoms), (**h**) MXene-5O-5Cl (5 oxygen and 5 chlorine atoms), (**i**) MXene-5OH-5F (5 fluorine atoms and 5 hydroxyl groups), (**j**) MXene-5OH-5Cl (5 chlorine atoms and 5 hydroxyl groups), (**k**) MXene-5F-5Cl (5 fluorine and 5 chlorine atoms).

**Figure 2 nanomaterials-14-00454-f002:**
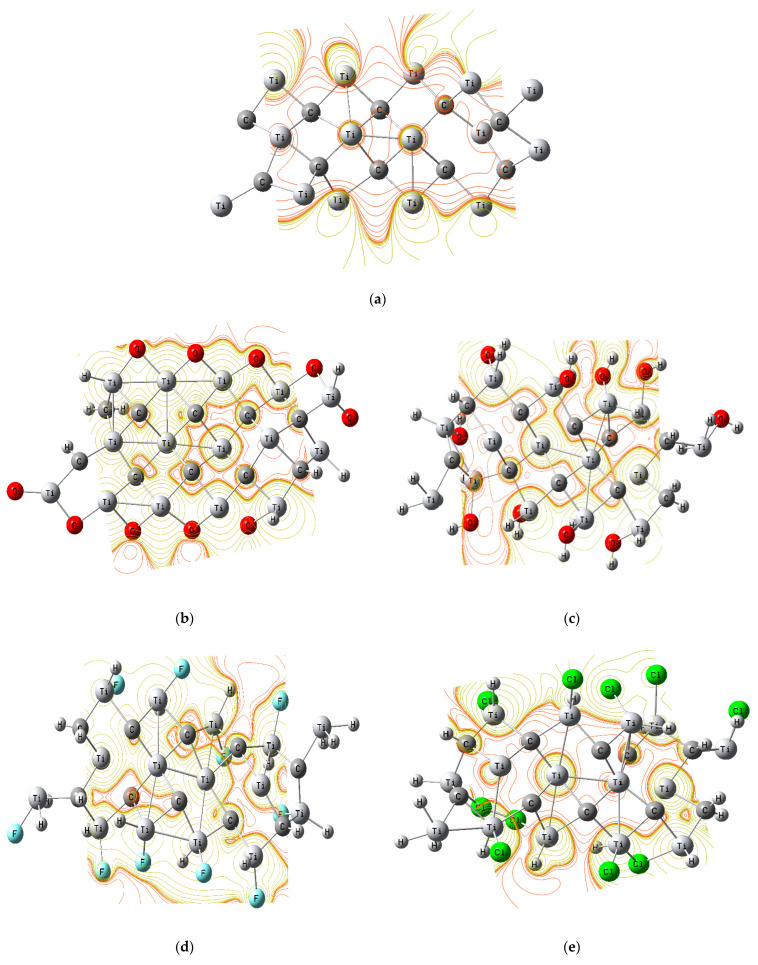
Molecular electrostatic potential (MESP) contour maps for two distinct molecular configurations depicting MXene (Ti_3_C_2_n__) and its functionalized counterpart (Ti_3_C_2_n__T_2_n__) with diverse functional groups (T=F, Cl, O, and OH), which are presented as follows: (**a**) pristine MXene, (**b**) MXene-10O (10 oxygen atoms), (**c**) MXene-10OH (10 hydroxyl groups), (**d**) MXene-10F (10 fluorine atoms), (**e**) MXene-10Cl (10 chlorine atoms), (**f**) MXene-5O-5OH (5 oxygen and 5 hydroxyl groups), (**g**) MXene-5O-5F (5 oxygen and fluorine atoms), (**h**) MXene-5O-5Cl (5 oxygen and 5 chlorine atoms), (**i**) MXene-5OH-5F (5 fluorine atoms and 5 hydroxyl groups), (**j**) MXene-5OH-5Cl (5 chlorine atoms and 5 hydroxyl groups), (**k**) MXene-5F-5Cl (5 fluorine and 5 chlorine atoms).

**Figure 3 nanomaterials-14-00454-f003:**
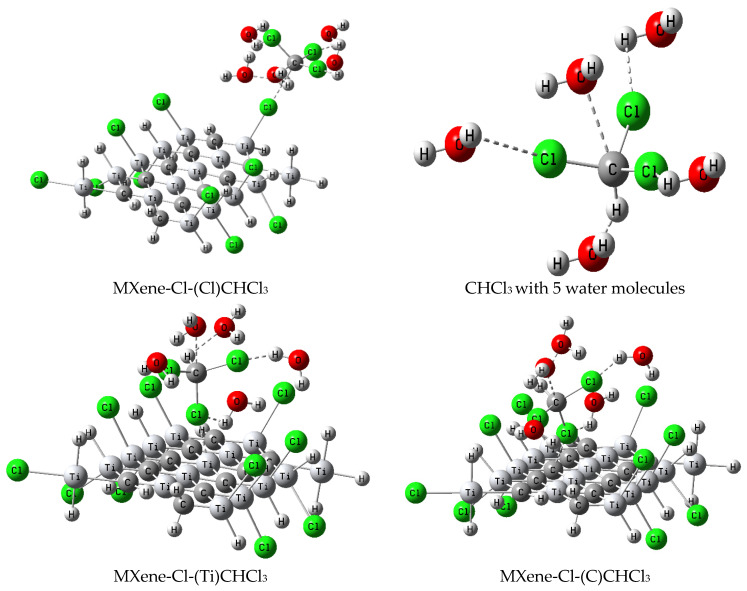
Optimized model structures illustrating the interaction of CHCl_3_ in water with active sites on functionalized MXene-Cl. The depictions emphasize CHCl_3_ interactions with specific active sites, corresponding to Cl (chlorine), Ti (titanium), and C (carbon) atoms in the functionalized MXene-Cl denoted as MXene-Cl-(Cl)CHCl_3_, MXene-Cl-(Ti)CHCl_3_, and MXene-Cl-(C)CHCl_3_, respectively.

**Figure 4 nanomaterials-14-00454-f004:**
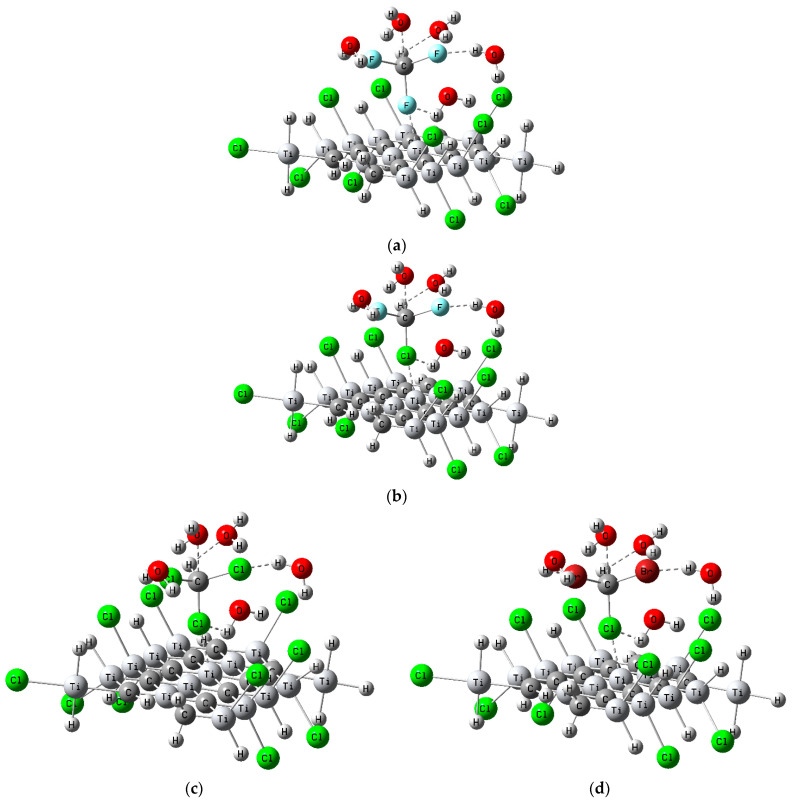
Simulated and optimized model structures of functionalized MXene-Cl designed for sensing common trihalomethanes (THMs), systematically ordered based on molecular weight. (**a**) CHF_3_: simulated optimized structure of MXene-Cl interacting with chloroform. (**b**) CHClF_2_: model structure for MXene-Cl with chlorodifluoromethane. (**c**) CHCl_3_: simulated optimized structure of MXene-Cl interacting with chloroform. (**d**) CHBrCl_2_: model structure for MXene-Cl with bromodichloromethane. (**e**) CHBr_2_Cl: simulated optimized structure of MXene-Cl with dibromochloromethane. (**f**) CHBr_3_: model structure for MXene-Cl with bromoform. (**g**) CHI_3_: simulated optimized structure of MXene-Cl interacting with iodoform. The ordering by molecular weight enables a coherent analysis of THM sensing interactions.

**Table 1 nanomaterials-14-00454-t001:** Calculated physical parameters including TDM (Debye) and ΔE (eV) of MXene Ti_3n_C_2n_ and functionalized MXene (Ti_3n_C_2n_T_2n_) with some functional groups (T=F, Cl, O, and OH), where n = 5 and using DFT: B3LYP/LANL2DZ model.

Structures	TDM (Debye)	ΔE (eV)
Mxene	6.399	0.633
Mxene-10O	10.927	0.855
Mxene-10OH	13.019	1.486
Mxene-10F	18.386	1.125
Mxene-10Cl	12.443	0.570
Mxene-5O-5OH	12.024	0.734
Mxene-5O-5F	13.213	0.940
Mxene-5O-5Cl	10.792	0.740
Mxene-5OH-5F	18.018	1.120
Mxene-5OH-5Cl	7.379	0.835
Mxene-5F-5Cl	13.569	0.927

**Table 2 nanomaterials-14-00454-t002:** Calculated approximated total energy and adsorption energy of functionalized MXene-Cl for CHCl_3_ sensing through different MXene-Cl active sites using DFT: B3LYP/LANL2DZ model.

Structure	TE (a.u)	E_Ads_ (a.u)
CHCl_3_	−465.319	-
MXene-Cl	−1411.995	-
MXene-Cl-(Ti)CHCl_3_	−1877.458	−0.145
MXene-Cl-(C)CHCl_3_	−1877.297	0.017
MXene-Cl-(Cl)CHCl_3_	−1866.977	10.337

**Table 3 nanomaterials-14-00454-t003:** DFT:B3LYP/LANL2DZ calculated TDM as (Debye) and ΔE (eV) for functionalized MXene-Cl for common THMs ordered by molecular weight.

Structures	TDM (Debye)	ΔE (eV)
Mxene-Cl	12.443	0.570
Mxene-Cl-CHF_3_	14.947	1.053
Mxene-Cl-CHClF_2_	13.680	0.887
Mxene-Cl-CHCl_3_	16.363	1.087
Mxene-Cl-CHBrCl_2_	15.998	0.685
Mxene-Cl-CHBr_2_Cl	16.017	0.665
Mxene-Cl-CHBr_3_	10.819	1.098
Mxene-Cl-CHI_3_	12.060	0.849

## Data Availability

The datasets used and/or analyzed during the current study are available from the corresponding author on reasonable request. Contact corresponding author: maryamg@mit.edu.

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
