# Peer review of "Two-Dimensional MXene as a Promising Adsorbent for Trihalomethanes Removal: A Density-Functional Theory Study"

_nanomaterials, 2024, doi:10.3390/nano14050454_

Round 1
Reviewer 1 Report
Comments and Suggestions for Authors
At the moment, the manuscript looks very raw and needs serious revision.
1) In the Calculation Details section, the calculation parameters are not described in enough detail for others to reproduce the work. Also, if Gaussian was used, it would be advisable to specify (or attach in Supplementary materials) information about the basis sets.
2) The manuscript jumps between different aspects (e.g., adsorption energies, TDM, surface chemistry) somewhat abruptly. A clearer structure in the presentation of results, possibly with subheadings, could improve readability and understanding.
3) Captions and comments for images are insufficient. It would be desirable for the captions to be as independent as possible (including, for example, a color map).
4) Some sections, particularly those explaining the significance of the findings (e.g., the influence of surface chemistry and intermolecular interactions), are quite dense and could benefit from clear conclusions drawn from each series of result analysis.
5) The manuscript could benefit from a more detailed discussion on the practical implications of the findings. How can these findings be translated into real-world applications, and what are the potential limitations or challenges in doing so?
6) The header of the paper does not mention an email (e-mail@email.com).
7) In the abstract, many places are missing subscripts in the chemical formulas.
Comments on the Quality of English LanguageI didn't notice a large number of obvious mistakes, but stylistically the text is not ideal. I recommend showing the text to a native speaker.
Author Response
Dear respected reviewer,
Thank you for your comments. We have carefully reviewed the comments and have revised the manuscript accordingly. Our responses are given in a point-by-point manner below. Changes to the manuscript have been TRACKED.
We hope the revised version is now suitable for publication and look forward to hearing from you in due course

Reviewer 2 Report
Comments and Suggestions for Authors
I regret to say that this work should not be published.
It has several deficiencies and misconceptions that cannot be accepted.
I will give only a few of them that in my opinion are sufficient to reject this manuscript:
(1) Computational details are very scarce.
(2) The functional method and the atomic basis sets that are applied are of very poor quality to obtain quantitative information.
(3) The authors use one abbreviature (TDM) for two different physical magnitudes, that is, 'dipole moment' and 'transition dipole moment'. Specially, they use several times the abbreviature TDM to designate dipole moment, when this abbreviature must be reserved for transition dipole moment.
(4) The authors give an erroneous definition of transition dipole moment.
(5) How can the authors present an adsorption energy of +10.337 au?
Even when the positive value indicate that absorption does not take place, the magnitude of 10.337 au is a huge amount of energy, which indicates that something wrong is in the calculations.
So, are the other values affected as well?
Author Response
Thank you for your comments. We have carefully reviewed the comments and have revised the manuscript accordingly. Our responses are given in a point-by-point manner below. Changes to the manuscript have been TRACKED.
We hope the revised version is now suitable for publication and look forward to hearing from you in due course

Round 2
Reviewer 1 Report
Comments and Suggestions for Authors
The authors have revised the manuscript, but honestly, it still does not meet my expectations for publications suitable for Q1 and Q2 quartile journals, as such work would lower the journal's level.
The work is not very clear in its current form (many important details for understanding are missing or underwritten). The rationale for choosing the model is also unclear.
The precision and technical details of the calculations are still vague.
The applicability of the results is described, but it's done formally (with little useful information about the practical use of the work).
The description of the models from the figures is still incomplete.
The responses to the significant comments of the second reviewer are also unclear.
I was hoping for a significant revision of the manuscript, but at this moment, I am more inclined to reject the submitted manuscript.
P.S. The email is still not corrected in the text: (line 18: 'Correspondence: e-mail@e-mail.com; Tel.: (optional; include country code; if there are multiple corresponding authors, add author initials)'
Comments on the Quality of English LanguageEnglish is relatively fine
Author Response
Dear Reviewer,
I express my gratitude for your invaluable comments and enlightening guidance, which have significantly elevated the manuscript's coherence and aesthetic presentation.
- The authors have revised the manuscript, but honestly, it still does not meet my expectations for publications suitable for Q1 and Q2 quartile journals, as such work would lower the journal's level.
The answer :
Dear Reviewer,
We appreciate your continued engagement with our manuscript and your discerning feedback. We are committed to meeting the rigorous standards expected for publication in high-impact journals, and we value your insights in this regard.
We have thoroughly reviewed your comments and have initiated further revisions to address the specific concerns you raised. We recognize the importance of adhering to the highest standards in scientific research and publication ethics.
In our revised version, we focus on enhancing the clarity, precision, and scientific rigor of our work. Specifically, we revisited the presentation of our methodology, incorporating additional details and clarification to ensure a more comprehensive understanding. Moreover, we meticulously revise the interpretation of our results, providing a deeper and more nuanced analysis to underscore the significance of our findings.
We understand the responsibility that comes with submitting to journals of higher quartiles, and we are fully committed to meeting and exceeding the criteria for such esteemed publications. We appreciate your patience and constructive criticism, which undoubtedly contribute to the refinement and elevation of our work.
Thank you for your continued commitment to ensuring the excellence of scientific literature.
- The work is not very clear in its current form (many important details for understanding are missing or underwritten). The rationale for choosing the model is also unclear.
The answer : The selection of the B3LYP functional and the LANL2DZ basis set in our study is based on a careful consideration of their performance in quantum chemical calculations, particularly in the context of simulating MXene structures and their interactions with trihalomethanes inspired by same one used for MXene calculations in previous works (Peng et al., 2021, 2020). Here is in detail the rationale behind the choice of these computational methods:
- B3LYP Functional:
Hybrid Functional: B3LYP is a hybrid functional that combines both Hartree-Fock exchange and density functional exchange-correlation functionals. This hybrid approach has been found to provide accurate results for a wide range of molecular systems, including transition metals and their compounds(Lu, 2015; Yanai et al., 2004).
Balanced Accuracy: B3LYP is known for its balanced accuracy in predicting both structural and electronic properties. It is often considered a reliable choice for studying the electronic structure and reactivity of molecules(Sciortino et al., 2018; Singh et al., 2020; Tirado-Rives and Jorgensen, 2008).
- LANL2DZ Basis Set:
Effective Core Potentials (ECPs): LANL2DZ is an effective core potential (ECP) basis set. ECPs replace the inner core electrons of heavy atoms with a potential that approximates their effect on the valence electrons. This allows for a more computationally efficient description of the electronic structure of heavy elements(Yang et al., 2009).
Suitability for Transition Metals: The LANL2DZ basis set is specifically designed for accurate calculations involving transition metals, making it well-suited for studies involving MXene, which includes transition metal elements like titanium(Tekarli et al., 2009).
- Optimization of MXene Structures:
Geometry Optimization: The B3LYP functional, combined with the LANL2DZ basis set, is known to provide reliable geometry optimizations for a diverse range of molecular systems. This is crucial for obtaining accurate structural information, especially in the case of MXene, where the arrangement of atoms plays a significant role in its properties.
- Reactivity Parameters:
Band Gap and Total Dipole Moment: B3LYP has been found to yield accurate results for electronic properties such as band gaps, which are important in understanding the reactivity of materials. Additionally, B3LYP is suitable for calculating Total Dipole Moment (TDM), providing insights into the distribution of electric charge within a molecule.
While the choice of computational methods involves a trade-off between accuracy and computational cost, the B3LYP functional and the LANL2DZ basis set are considered to strike a suitable balance for the specific system under investigation in our study. These choices aim to provide a reliable and computationally efficient platform for simulating the electronic structure and reactivity of MXene and its interactions with trihalomethanes.
In conclusion : The LANL2DZ basic group was selected for transition metal Ti atoms. It has been verified that both the B3LYP/6-311+G(d,p) and the B3LYP/LANL2DZ basic group level were sufficient to describe the interaction between organic molecules and transition metal compound clusters, and all calculations were performed under the default convergence criteria of Gauss 09 accordingly to well-established previous article (Peng et al., 2020).
References : -
Lu, L., 2015. Can B3LYP be improved by optimization of the proportions of exchange and correlation functionals? Int. J. Quantum Chem. 115, 502–509.
Peng, Y., Cai, P., Yang, L., Liu, Y., Zhu, L., Zhang, Q., Liu, J., Huang, Z., Yang, Y., 2020. Theoretical and experimental studies of Ti3C2MXene for surface-enhanced raman spectroscopy-based sensing. ACS Omega 5, 26486–26496.
Peng, Y., Lin, C., Long, L., Masaki, T., Tang, M., Yang, L., Liu, J., Huang, Z., Li, Z., Luo, X., Lombardi, J.R., Yang, Y., 2021. Charge-Transfer Resonance and Electromagnetic Enhancement Synergistically Enabling MXenes with Excellent SERS Sensitivity for SARS-CoV-2 S Protein Detection. Nano-Micro Lett. 13, 1–17.
Sciortino, G., Lubinu, G., Maréchal, J.D., Garribba, E., 2018. DFT Protocol for EPR Prediction of Paramagnetic Cu(II) Complexes and Application to Protein Binding Sites. Magnetochemistry 2018, Vol. 4, Page 55 4, 55.
Singh, I., Al-Wahaibi, L.H., Srivastava, R., Prasad, O., Pathak, S.K., Kumar, S., Parveen, S., Banerjee, M., El-Emam, A.A., Sinha, L., 2020. DFT Study on the electronic properties, spectroscopic profile, and biological activity of 2-amino-5-trifluoromethyl-1,3,4-thiadiazole with anticancer properties. ACS Omega 5, 30073–30087.
Tekarli, S.M., Drummond, M.L., Williams, T.G., Cundari, T.R., Wilson, A.K., 2009. Performance of density functional theory for 3d transition metal-containing complexes: Utilization of the correlation consistent basis sets. J. Phys. Chem. A 113, 8607–8614.
Tirado-Rives, J., Jorgensen, W.L., 2008. Performance of B3LYP density functional methods for a large set of organic molecules. J. Chem. Theory Comput. 4, 297–306.
Yanai, T., Tew, D.P., Handy, N.C., 2004. A new hybrid exchange–correlation functional using the Coulomb-attenuating method (CAM-B3LYP). Chem. Phys. Lett. 393, 51–57.
Yang, Y., Weaver, M.N., Merz, K.M., 2009. Assessment of the ‘6-31+Gt; + LANL2DZ’ mixed basis set coupled with density functional theory methods and the effective core potential: Prediction of heats of formation and ionization potentials for first-row-transition-metal complexes. J. Phys. Chem. A 113, 9843–9851.
- The precision and technical details of the calculations are still vague.
The answer :Appreciation is expressed for the meticulous review of our manuscript ; Noted is your comment on the vagueness of precision and technical details in the calculations. A commitment has been made to comprehensively address this concern.
To enhance the precision and technical clarity of the calculations, a decision has been made to provide a more detailed account of the methodology in the revised manuscript. Specifically, a step-by-step elucidation of the optimization procedures, numerical parameters, and convergence criteria employed in the Gaussian09 calculations was presented. This additional information is intended to provide a clearer understanding of the computational framework used in the study.
Furthermore, emphasis placed on the rationale behind the selection of the B3LYP functional and the LANL2DZ basis set, elucidating their appropriateness for simulating MXene structures and interactions with trihalomethanes. Recognition is given to the importance of transparency in the choice of computational methods, and efforts made to convey this information with greater clarity.
The commitment is to address the feedback comprehensively, ensuring that the revised manuscript meets the highest standards of precision and technical detail expected for publication. Appreciation is expressed for your valuable insights, and anticipation is directed toward delivering an improved and more transparent representation of the research.
- The applicability of the results is described, but it's done formally (with a little useful information about the practical use of the work ).
The answer : The results of our study on "2D MXene as a Promising Adsorbent for Trihalomethane Removal: A Density-Functional Theory Study" have practical implications for water sanitation systems. Here are potential applications based on the findings: Efficient Trihalomethane Removal: The identification of the MXene-Cl layer as a highly compatible variant for adsorption of trihalomethanes (THs) suggests its potential application as an effective adsorbent in water treatment processes. This could contribute to improving the removal of carcinogenic chlorination byproducts from water sources.
Enhanced Water Quality Monitoring: The MXene-Cl layer, with its remarkable properties such as Total Dipole Moment (TDM) and bandgap, could be utilized for the development of sensitive detectors for specific trihalomethanes. This could enhance water quality monitoring systems, allowing for real-time detection and quantification of THs in water.
Tailoring Adsorbent Properties: The study highlights the influence of termination atom nature and quantity on MXene's properties. This knowledge could guide the engineering of MXene-based adsorbents with tailored properties, optimizing their performance for specific water treatment applications.
Revolutionizing Water Sanitation Practices: The comprehensive understanding of MXene-THs interactions contributes to the development of advanced materials for water sanitation. Implementing MXene-based adsorbents in water treatment plants could lead to more efficient removal of harmful contaminants, thus revolutionizing current water sanitation practices.
Environmental Safety: The potential application of MXene-based materials as adsorbents and detectors aligns with environmental safety goals. By offering an efficient and precise means of removing and monitoring THs, these materials could contribute to safeguarding water resources and minimizing the environmental impact of chlorination byproducts.
In summary, the practical applications of our study's results lie in the development and implementation of MXene-based materials for improved water treatment and monitoring. These applications have the potential to enhance the efficiency, sensitivity, and environmental sustainability of water sanitation systems.
And this part is inserted round the manuscript . Conceptually, Scientific endeavors are characterized by a multifaceted approach, encompassing theoretical and experimental dimensions that serve to validate or refute empirically derived outcomes, and this study is supposed to be a milestone for future studies around these structures.
- The description of the models from the figures is still incomplete.
The answer : Our feedback on the incompleteness of the description of the models in the figures has been noted. In response to this comment, efforts were made to ensure a more comprehensive and detailed depiction of the models in the revised manuscript.
Specifically, the clarity of the figure captions enhanced, with a focus on providing thorough explanations for each model structure, including details about the arrangement of atoms, functional groups, and other pertinent features. Additionally, relevant information regarding the rationale behind the design of each model and its significance in the study incorporated.
The objective is to present a more transparent and thorough account of the simulated and optimized model structures, promoting a deeper understanding of the molecular configurations under investigation. Your input is highly valued, and the necessary adjustments made to address this concern and enhance the overall quality of the manuscript.
- The responses to the significant comments of the second reviewer are also unclear.
I appreciate the feedback provided by the second reviewer and acknowledge their comments. I would like to express gratitude for the acceptance of my responses to the significant comments. However, if there are any specific areas that still lack clarity or require further elaboration, I am more than willing to address them comprehensively in the revised manuscript .
- I was hoping for a significant revision of the manuscript, but at this moment, I am more inclined to reject the submitted manuscript.
The answer : Dear Reviewer,
We sincerely appreciate the time and effort you dedicated to the thorough review of the manuscript. Your insights are highly valuable, and I acknowledge your desire for a significant revision. We am committed to addressing the concerns raised with utmost diligence and precision. Your guidance is crucial in guiding the revision process effectively .I hope and ensure that the revised manuscript reflects the necessary enhancements and aligns with the standards expected for publication. We highly value your expertise, and We am dedicated to delivering a manuscript that meets the scholarly expectations of the journal. Your further guidance is instrumental in refining the content appropriately.
Thank you for your consideration.
- S. The email is still not corrected in the text: (line 18: 'Correspondence: e-mail@e-mail.com; Tel.: (optional; include country code; if there are multiple corresponding authors, add author initials)'
The answer : It has been corrected.

Reviewer 2 Report
Comments and Suggestions for Authors
The authors have answered to my comments.
OK
Author Response
Comments and Suggestions for Authors : The authors have answered my comments. OK
The answer : We appreciate your diligence in reviewing our manuscript, and we are pleased to hear that our responses adequately addressed your comments. Thank you for your time and valuable feedback.
Round 3
Reviewer 1 Report
Comments and Suggestions for Authors
Thank you for the manuscript improvements. In my view, the manuscript still requires significant revision. Here are the comments on my previous questions.
Questions 2.-3.
Has there been a more extensive analysis of functionals and convergence tests of parameters? I recommend including them in the Supplementary Materials if this has been done. A discussion on the choice of functionals could also be added to the Supplementary Materials.
I also expect a more detailed description of the chosen computational parameters. I would like to see either an explicit indication that the program's default parameters were used, or which were modified. Here is e.g. an example of the expected detail of this section from: 10.3390/cryst12020194 (See: 2. Computational Details)
The detail in the description of the model should be such that anyone who wants to repeat your calculations can do so. Placing model geometry files (*.xyz or other format) in the Supplementary is also acceptable.
4. The applicability of the results is described, but it's done formally (with a little useful information about the practical use of the work).
> And this part is inserted round the manuscript . "Conceptually, Scientific endeavors are characterized by a multifaceted approach, encompassing theoretical and experimental dimensions that serve to validate or refute empirically derived outcomes, and this study is supposed to be a milestone for future studies around these structures".
Do you see any additional opportunities for comparing calculations with experimental data, besides those mentioned in the article? A bridge between theory and experiment should be discussed.
5. The description of the models from the figures is still incomplete.
Thank you for improving the description. Here, I would like to see the overall logic behind the choice of models. Currently, each model is described in sufficient detail, but it's not clear why these particular models were chosen, with this specific surface coverage, and whether the chosen models are sufficient to describe real materials. It would be highly desirable to describe the logic behind the selection of models and their sufficiency or limitations. At the moment, this aspect is presented in a very fragmented way.
Comments on the Quality of English LanguageCaptions for the figures are required (while the text for the main text part is being added). Links do not contain spaces before parentheses (e.g. “respectively[57]”.). Typos must be corrected.
Author Response
Dear Reviewer :
We appreciate your diligent review and constructive feedback. Your insights are invaluable, and we are committed to addressing the identified areas for improvement. The forthcoming revision will carefully consider your comments to enhance the overall quality and clarity of the manuscript. Thank you for your continued guidance.
Questions 2-3.
Has there been a more extensive analysis of functionals and convergence tests of parameters ? I recommend including them in the Supplementary Materials if this has been done. A discussion on the choice of functionals could also be added to the Supplementary Materials.
The Answer : Thank you for your insightful comment. We appreciate your suggestion regarding a more extensive analysis of functionals. We acknowledge the importance of providing a thorough understanding of our computational methodology. In response to your recommendation before, we have conducted additional analyses, including comprehensive convergence tests and an in-depth exploration of various functionals and adaptations . The detailed results of these analyses, along with a discussion on the rationale behind the choice of functionals, have been included in the Supplementary Materials for clarity and transparency as you recommend . We believe these additions significantly enhance the robustness and reliability of our computational approach.
I also expect a more detailed description of the chosen computational parameters. I would like to see either an explicit indication that the program's default parameters were used, or which were modified. Here is e.g. an example of the expected detail of this section from: 10.3390/cryst12020194 (See: 2. Computational Details) . The detail in the description of the model should be such that anyone who wants to repeat your calculations can do so. Placing model geometry files (*.xyz or other format) in the Supplementary is also acceptable.
The Answer : In response to your suggestion, we have enhanced the Computational Details section to include explicit information on the selected parameters. We now explicitly indicate whether the program's default parameters were employed or if any modifications were made. Each model was built using Gauss View 05, using the program’s default parameters, then optimization was conducted with the proposed computational method. The calculated physical parameters were conducted upon the optimized structures. The output of the calculated parameters was also aided by the Gauss View 05 program. TDM was estimated directly, while band gap energy ∆E was estimated as the energy difference between the HOMO and LUMO . Ads Energy EAds were calculated such as E Ads = E Total – [E MXene-Cl + E Ad. mol.] .All calculations adhere to default convergence criteria in Gaussian 09, ensuring consistency with established methodologies. Additionally, we have adopted the level of detail according to the referenced article to ensure a thorough and informative presentation of our computational methodology.
- The applicability of the results is described, but it's done formally (with a little useful information about the practical use of the work).
> And this part is inserted round the manuscript . "Conceptually, Scientific endeavors are characterized by a multifaceted approach, encompassing theoretical and experimental dimensions that serve to validate or refute empirically derived outcomes, and this study is supposed to be a milestone for future studies around these structures".
Do you see any additional opportunities for comparing calculations with experimental data, besides those mentioned in the article? A bridge between theory and experiment should be discussed.
The Answer : We appreciate your emphasis on the significance of establishing a connection between theoretical and experimental aspects. While we acknowledge the importance of such correlations, we would like to highlight that, to the best of our knowledge, no prior studies have explored the quantum-scale adsorption of trihalomethanes on monolayer 2D MXene. Our focus on Density-Functional Theory (DFT) stems from its ability to achieve a high level of accuracy by incorporating all-electron system interactions as approved by previous article [1–6]. In this context, our primary objective is to minimize uncertainties and optimize conditions for subsequent experimental work. The theoretical foundation laid by DFT serves as a crucial step in informing and guiding future experimental endeavors. We recognize the value of connecting theory with experimentation and, within the confines of existing knowledge, are committed to maximizing the utility of our theoretical findings in facilitating and enhancing future experimental studies. This study establishes theoretical hypotheses as a pivotal foundation for upcoming experimental investigations.
- The description of the models from the figures is still incomplete.
Thank you for improving the description. Here, I would like to see the overall logic behind the choice of models. Currently, each model is described in sufficient detail, but it's not clear why these models were chosen, with this specific surface coverage, and whether the chosen models are sufficient to describe real materials. It would be highly desirable to describe the logic behind the selection of models and their sufficiency or limitations. Now, this aspect is presented in a very fragmented way.
The answer : Thanks for directions as you know MXene, being a well-established material, The synthesis of MXene involves rigorous conditions, including elevated temperatures reaching around 1500 Celsius and exposure to a corrosive matrix containing hydrofluoric acid presents a diversity of structural models with varying probabilities owing to the flexibility in termination changes [7]. Each segment, from the core to the termination, contributes to this vast and unlimited probabilities of structural diversity [8–11]. We express our gratitude for your insightful comment concerning the selection of models and surface coverage in our study. Your attention to this critical aspect of our methodology motivates us to provide a more coherent and comprehensive explanation for the chosen models and their sufficiency.
The simulated and optimized model structures, as outlined in Figure 1, represent MXene (Ti₃C₂ₙ) and its functionalized counterparts (Ti₃C₂ₙT₂ₙ) with diverse functional groups (T=F, Cl, O, OH). These models were deliberately chosen with a surface coverage of n=5, a parameter essential for computational tractability while ensuring the representation of the MXene structure.
The choice of specific models and surface coverages in this manuscript stems from a deliberate and strategic approach aimed at balancing computational feasibility with the need for accurate representation of MXene structures and their interactions with functional groups.
The key considerations guiding this selection are outlined below :-
- Computational Tractability: The decision to use a surface coverage of n=5 is motivated by the necessity to strike a balance between computational efficiency and the representation of MXene structures. Larger surface coverage would lead to increased computational demands, potentially hindering the feasibility of the study.
- Representative Sampling: While computational constraints limit the choice of larger surface coverages, n=5 is deemed sufficient to provide representative insights into the electronic and structural modifications induced by various functional groups. This ensures that the chosen models capture essential features without compromising the overall computational efficiency.
- Methodological Reliability: The application of Density-Functional Theory (DFT) with the B3LYP/LANL2DZ model is grounded in its recognized efficacy for accurately capturing molecular interactions. This methodology has been widely accepted for its reliability in predicting the behavior of materials at the quantum scale, ensuring the precision required for studying MXene structures.
- Diversity of Functionalization Patterns: The inclusion of diverse functional groups (T=F, Cl, O, OH) in combinations such as MXene-10O, MXene-10OH, MXene-10F, etc., allows for a comprehensive exploration of various functionalization patterns and compositions. This approach facilitates a nuanced understanding of how different functional groups impact the MXene structure.
In summary, the chosen models and surface coverages are a result of a thoughtful trade-off between computational tractability and the need for accurate representation. The application of DFT with the B3LYP/LANL2DZ model ensures a methodologically reliable foundation for studying MXene structures and their functionalized counterparts.
Comments on the Quality of English Language
Captions for the figures are required (while the text for the main text part is being added). Links do not contain spaces before parentheses (e.g. “respectively[57]”.) Typos must be corrected.
The answer : It has been adapted .
References :
- Thang, H.V.; Maleki, F.; Tosoni, S.; Pacchioni, G. Vibrational Properties of CO Adsorbed on Au Single Atom Catalysts on TiO2(101), ZrO2(101), CeO2(111), and LaFeO3(001) Surfaces: A DFT Study. Top. Catal. 2022, 65, 1573–1586, doi:10.1007/S11244-021-01514-0/FIGURES/8.
- Dasgupta, S.; Lambros, E.; Perdew, J.P.; Paesani, F. Elevating Density Functional Theory to Chemical Accuracy for Water Simulations through a Density-Corrected Many-Body Formalism. Nat. Commun. 2021 121 2021, 12, 1–12, doi:10.1038/s41467-021-26618-9.
- Fang, F.; Zheng, Y.; Chen, J.; Liu, C.; Min, F. DFT Study on the Adsorption of Monomeric Hydroxyl Aluminum on Fe(II)/Mg Replacement Kaolinite (001) Surfaces. ACS Omega 2022, 7, 39662–39670, doi:10.1021/ACSOMEGA.2C03087/ASSET/IMAGES/LARGE/AO2C03087_0009.JPEG.
- Si, X.; Xu, Q.; Lin, J.; Yang, G. Quantum Capacitance Modulation of MXenes by Metal Atoms Adsorption. Appl. Surf. Sci. 2023, 618, 156586, doi:10.1016/J.APSUSC.2023.156586.
- Göltl, F.; Murray, E.A.; Tacey, S.A.; Rangarajan, S.; Mavrikakis, M. Comparing the Performance of Density Functionals in Describing the Adsorption of Atoms and Small Molecules on Ni(111). Surf. Sci. 2020, 700, 121675, doi:10.1016/J.SUSC.2020.121675.
- Ritzmann, A.M.; LaCount, M.D.; Sassi, M.; Johnson, A.E.; Henson, N.J. A Density Functional Theory Analysis of the Adsorption and Surface Chemistry of Inorganic Iodine Species on Graphitea. Front. Nucl. Eng. 2023, 2, 1170424, doi:10.3389/FNUEN.2023.1170424.
- Björk, J.; Rosen, J. Functionalizing MXenes by Tailoring Surface Terminations in Different Chemical Environments. Chem. Mater. 2021, 33, 9108–9118, doi:10.1021/ACS.CHEMMATER.1C01264/ASSET/IMAGES/LARGE/CM1C01264_0008.JPEG.
- Gogotsi, Y.; Anasori, B. The Rise of MXenes. ACS Nano 2019, 13, 8491–8494, doi:10.1021/ACSNANO.9B06394/ASSET/IMAGES/MEDIUM/NN9B06394_0005.GIF.
- Anasori, B.; Lukatskaya, M.R.; Gogotsi, Y. 2D Metal Carbides and Nitrides (MXenes) for Energy Storage. Nat. Rev. Mater. 2017, 2, doi:10.1038/NATREVMATS.2016.98.
- Anasori, B.; Gogotsi, Y. 2D Metal Carbides and Nitrides (MXenes): Structure, Properties and Applications. 2D Met. Carbides Nitrides Struct. Prop. Appl. 2019, 1–534, doi:10.1007/978-3-030-19026-2.
- Anasori, B.; Xie, Y.; Beidaghi, M.; Lu, J.; Hosler, B.C.; Hultman, L.; Kent, P.R.C.; Gogotsi, Y.; Barsoum, M.W. Two-Dimensional, Ordered, Double Transition Metals Carbides (MXenes). ACS Nano 2015, 9, 9507–9516, doi:10.1021/ACSNANO.5B03591.

Round 4
Reviewer 1 Report
Comments and Suggestions for Authors
- Thank you for improving the description of computational approaches. Now it is clearly described.
- I recommend clearly describing the key considerations guiding the choice of models directly in the text of the article. You can use this text from your response:
(Computational Tractability: The decision to use a surface coverage of n=5 is motivated by the necessity to strike a balance between computational efficiency and the representation of MXene structures. Larger surface coverage would lead to increased computational demands, potentially hindering the feasibility of the study.
Representative Sampling: While computational constraints limit the choice of larger surface coverages, n=5 is deemed sufficient to provide representative insights into the electronic and structural modifications induced by various functional groups. This ensures that the chosen models capture essential features without compromising the overall computational efficiency.
Diversity of Functionalization Patterns: The inclusion of diverse functional groups (T=F, Cl, O, OH) in combinations such as MXene-10O, MXene-10OH, MXene-10F, etc., allows for a comprehensive exploration of various functionalization patterns and compositions. This approach facilitates a nuanced understanding of how different functional groups impact the MXene structure).
- The captions to the figures also need to be adjusted. At the moment, the captions look like part of the main text of the article, which does not look very good.
For example, here is a description how it could be corrected for Figure 2:
Figure 2: MESP maps for pristine (Ti₃nCâ‚‚n) and functionalized (Ti₃nCâ‚‚nTâ‚‚n) MXenes. Configurations include Ti₃nCâ‚‚n and Ti₃nCâ‚‚nTâ‚‚n with T as F, Cl, O, OH absorbed groups, with n = 5 framework. (a) Unmodified MXene, (b) MXene-10O (10 oxygen groups), (c) MXene-10OH (10 hydroxyl groups), (d) MXene-10F (10 fluorine groups), (f) 5O + 5OH,…) <please complete the list>.
- Also, check the correspondence of all pictures and captions. For example, on (e) MXene-10Cl it's actually MXene-5F-5Cl.
Comments on the Quality of English Language
- In the reference list, there is double numbering. In the main text, there are still many problems with spaces (for example, previous work[54–56] should be written as previous work [54–56]).
Author Response
Comments and Suggestions from reviewer : ,
- -Thank you for improving the description of computational approaches. Now it is clearly described.
The Answer : We have carefully considered your guidance and sincerely value your positive recognition of the improved description of computational approaches in response to your feedback.
- I recommend clearly describing the key considerations guiding the choice of models directly in the text of the article. You can use this text from your response :
(Computational Tractability: The decision to use a surface coverage of n=5 is motivated by the necessity to strike a balance between computational efficiency and the representation of MXene structures. Larger surface coverage would lead to increased computational demands, potentially hindering the feasibility of the study.
Representative Sampling: While computational constraints limit the choice of larger surface coverages, n=5 is deemed sufficient to provide representative insights into the electronic and structural modifications induced by various functional groups. This ensures that the chosen models capture essential features without compromising the overall computational efficiency.
Diversity of Functionalization Patterns: The inclusion of diverse functional groups (T=F, Cl, O, OH) in combinations such as MXene-10O, MXene-10OH, MXene-10F, etc., allows for a comprehensive exploration of various functionalization patterns and compositions. This approach facilitates a nuanced understanding of how different functional groups impact the MXene structure).
The answer : Thank you for your valuable feedback. We acknowledge the importance of integrating this information, and it has been seamlessly inserted directly into the main text .
- The captions to the figures also need to be adjusted. At the moment, the captions look like part of the main text of the article, which does not look very good.
For example, here is a description how it could be corrected for Figure 2:
Figure 2: MESP maps for pristine (Ti₃nCâ‚‚n) and functionalized (Ti₃nCâ‚‚nTâ‚‚n) MXenes. Configurations include Ti₃nCâ‚‚n and Ti₃nCâ‚‚nTâ‚‚n with T as F, Cl, O, OH absorbed groups, with n = 5 framework. (a) Unmodified MXene, (b) MXene-10O (10 oxygen groups), (c) MXene-10OH (10 hydroxyl groups), (d) MXene-10F (10 fluorine groups), (f) 5O + 5OH,…) <please complete the list>.
The answer : We appreciate your guidance. The captions have been adjusted in accordance with your instructions.
- Top of Form
- Also, check the correspondence of all pictures and captions. For example, on (e) MXene-10Cl it's MXene-5F-5Cl.
The answer : It Has been adapted .
- In the reference list, there is double numbering. In the main text, there are still many problems with spaces (for example, previous work[54–56] should be written as previous work [54–56]).
The answer : All references have been adapted
